# Identification of a Quinone Derivative as a YAP/TEAD Activity Modulator from a Repurposing Library

**DOI:** 10.3390/pharmaceutics14020391

**Published:** 2022-02-10

**Authors:** Angela Lauriola, Elisa Uliassi, Matteo Santucci, Maria Laura Bolognesi, Marco Mor, Laura Scalvini, Gian Marco Elisi, Gaia Gozzi, Lorenzo Tagliazucchi, Gaetano Marverti, Stefania Ferrari, Lorena Losi, Domenico D’Arca, Maria Paola Costi

**Affiliations:** 1Department of Life Sciences, University of Modena and Reggio Emilia, Via G. Campi 103, 41125 Modena, Italy; angela.lauriola@univr.it (A.L.); matteo.santucci@univr.it (M.S.); lorenzo.tagliazucchi@unimore.it (L.T.); sferrari591@gmail.com (S.F.); lorena.losi@unimore.it (L.L.); 2Department of Pharmacy and Biotechnology, Alma Mater Studiorum—University of Bologna, Via Belmeloro 6, 40126 Bologna, Italy; elisa.uliassi3@unibo.it (E.U.); marialaura.bolognesi@unibo.it (M.L.B.); 3Dipartimento di Scienze degli Alimenti e del Farmaco, Università degli Studi di Parma, Parco Area delle Scienze 27/A, I, 43124 Parma, Italy; marco.mor@unipr.it (M.M.); laura.scalvini@unipr.it (L.S.); gianmarco.elisi@unipr.it (G.M.E.); 4Department of Biomedical, Metabolic and Neural Sciences, University of Modena and Reggio Emilia, Via G. Campi 287, 41125 Modena, Italy; g.gozzi@holostem.com (G.G.); gaetano.marverti@unimore.it (G.M.)

**Keywords:** TEAD transcription factor, leads repurposing, luciferase assay, chemoinformatic, p-quinoid derivatives

## Abstract

The transcriptional regulators YAP (Yes-associated protein) and TAZ (transcriptional co-activator with PDZ-binding motif) are the major downstream effectors in the Hippo pathway and are involved in cancer progression through modulation of the activity of TEAD (transcriptional enhanced associate domain) transcription factors. To exploit the advantages of drug repurposing in the search of new drugs, we developed a similar approach for the identification of new hits interfering with TEAD target gene expression. In our study, a 27-member in-house library was assembled, characterized, and screened for its cancer cell growth inhibition effect. In a secondary luciferase-based assay, only seven compounds confirmed their specific involvement in TEAD activity. **IA5** bearing a p-quinoid structure reduced the cytoplasmic level of phosphorylated YAP and the YAP–TEAD complex transcriptional activity and reduced cancer cell growth. **IA5** is a promising hit compound for TEAD activity modulator development.

## 1. Introduction

Yes1-associated protein (YAP) and transcriptional co-activator with PDZ-binding motif (TAZ) are two oncogenic transcriptional co-activators that bind the transcriptional enhanced associate domain (TEAD) transcription factors and are, thus, involved in cell proliferation. In fact, the YAP and TAZ functional activity is implicated in the tumorigenesis of various cancers, including breast [1,2], colon [3], and lung [4]. Indeed, in tumors, these proteins can reprogram cancer cells into cancer stem cells and incite tumor initiation, progression, and metastasis [5,6,7]. They are strictly regulated by the Hippo kinase cascade, whose output effect consists in inhibiting their nuclear translocation and responsive target gene transcription [6,8]. As shown in Figure 1, the kinasic core-module of the Hippo pathway, including MST1/2 (STK4, Serine/Threonine Kinase 4), Mob1A/1B (MOB kinase activator 1(A/B), and SAV1 (Salvador Family WW Domain Containing Protein 1), induces LATS1/2 (Large tumor suppressor kinase 1)-mediated phosphorylation of YAP with its consequent cytoplasmic retention (Ser127) or destruction by proteasomal degradation (Ser381) [9]. In addition, several cell junction proteins are able to bind YAP by direct protein–protein interactions (PPIs), preventing its translocation to the nucleus, and then regulating the YAP subcellular localization in a phosphorylation-independent manner [10]. AMOT family proteins and α-catenin are two important components of tight- (TJs) and adherent-junctions (AJs), respectively. An interaction between AMOT and YAP/TAZ has been identified, and it is not dependent on the YAP/TAZ phosphorylation status, consequently resulting in reduced YAP/TAZ nuclear localization and activity. Unlike AMOT, YAP/α-catenin binding is based on the YAP-phosphorylation at Ser127; in this way, the 14-3-3 complex can interact with phosphorylated YAP (pYAP) and then with α-catenin protein, to sequester YAP at AJs and prevent its dephosphorylation/activation. Unlike the phosphorylation on Ser127, YAP phosphorylated on Ser381 undergoes a proteasomal degradation through the recruitment of SCFβ-TRCP ubiquitin E3 ligase complex, with a consequent decrease of YAP protein levels. Based on the previous considerations, inhibiting YAP/TAZ–TEAD complex represents an attractive and viable strategy to restrain the transcriptional outputs of YAP/TAZ for novel cancer therapies [11,12,13].

Current inhibitors of YAP–TEAD transcriptional activity may act by directly altering the YAP–TEAD interaction or by indirectly interfering with the complex transcriptional activity through modulation of the multiple activators [13]. This can happen following different mechanisms, summarized in a sunburst graph and detailed description in Appendix A. All these compounds have the same common output, to directly or indirectly block YAP–TEAD activity [14]. Among the direct inhibitors, verteporfin (VP), an FDA-approved photosensitizing compound, has been reported as a TEAD gene transcription disrupter [15,16,17]. However, its mechanism of action is not yet fully clarified. Similar to VP, the pentacyclic terpenoid, celastrol, blocked YAP–TEAD binding, leading to the inhibition of TEAD-dependent target gene expression in cancer cell lines [18]. Additionally, peptides derived from both the YAP Ω-loop sequence (YAP cyclic peptide 17 and TB1G1 cystine-dense peptide) [19,20] and the co-regulator Vgll4, [21] a structurally YAP/TAZ mutually-exclusive competitor for TEAD binding, were shown to act as protein–protein interaction (PPI) disruptors. Small molecules have been also designed to target the TEAD surface interacting with the YAP Ω-loop [22]. Digitoxin was putatively identified as a YAP WW domain modulator by in silico analysis [23] and was found to promote YAP protein levels in the cytoplasm [24]. The palmitate-binding pocket (PBP) represents the most druggable pocket located at the center of the TEAD’s YAP/TAZ binding domain, and several small molecules have been proposed (flufenamic acid derivatives, DC-TEADin02 [25] and K-975 [26]). On the other hand, indirect inhibitors include structurally different compounds, targeting a great variety of proteins (e.g., epigenetic proteins, GPCRs, kinases), but with all ultimately acting as LATS protein phosphorylation enhancers (Appendix A). Phosphatases and intracellular kinases inhibitors (calyculin-A, dasatinib, and trametinib), as well as the inhibitors of cell surface receptors (glucagon/epinephrin/losartan and erlotinib/pazopanib/axitinib); the mevalonate pathway inhibitors (statins, zoledronic acid and GGTI-298); and the cytoskeletal actin-, epigenetic-, and cellular energy stress-modulators (blebbistatin/latrunculin-A/XAV-939, panobinostat/I-BET151 and metformin/AICAR) have been reported as strong modulators of the YAP–TEAD network by enhancing the LATS-dependent inhibitory phosphorylation of YAP/TAZ, to reduce their transcriptional outputs. Many of these drugs have the potential to be repurposed as YAP/TAZ indirect inhibitors, to treat various solid cancers [13]. 

With the aim of identifying and characterizing new compounds with lead-like properties that could modulate YAP-TEAD activity, in the present study, we followed a repurposing strategy, by selecting and screening a small library of 27 compounds. This library contains preclinical compounds contributed by our academic laboratories in different medicinal chemistry programs. Thus, most of the collection is comprised of lead-like molecules, which offer excellent opportunities for repurposing. A few commercially available compounds, including known modulators of the Hippo pathway, were acquired as an internal reference. Since the biological crosstalk with the YAP/TAZ–TEAD pathway is mediated by PPI [27] (Figure 1), the selected compounds have structural features for acting as PPI inhibitors (iPPI) and indirectly disrupt the YAP/TAZ–TEAD interaction (Figure 2). 

The library was constructed using chemioinformatic and bioinformatic tools, to characterize investigational candidates and drugs repurposed in comparison with known inhibitors. All compounds were tested and prioritized based on their inhibitory potency in a HaCaT cell proliferation assay, used as a cellular tool to identify those compounds able to affect cell growth [24]. The best cell growth inhibitors were progressively characterized in a secondary assay for their capacity to inhibit YAP-TEAD activity in a HaCaT cell luciferase assay and to display cytotoxic effects on several cancer cell lines. We finally ascertained that some selected compounds specifically interfere with the YAP phosphorylation and TEAD gene expression activity in two colorectal cancer cell lines, HCT116 and HT29.

## 2. Materials and Methods

### 2.1. Chemistry

All commercially available reagents (including pristimerin, digoxin, deslanoside, and verteporfin) and solvents were purchased from Sigma—Aldrich, Fluka (Italy), TCI, Alpha Aesar, without further purification. Reactions were followed by analytical thin layer chromatography (TLC), performed on precoated TLC plates (0.20 mm silica gel 60 with UV254 fluorescent indicator, Merck). Developed plates were air-dried and visualized by exposure to UV light (l = 254 nm and 365 nm). Reactions involving the generation or consumption of amine were visualized using bromocresol green spray (0.04% in EtOH, made blue by NaOH). Column chromatography purifications were performed under flash conditions using Kieselgel 40 (0.040–0.063 mm; Merck). NMR experiments for compound **1** were run on a Varian VXR400, while for compound **2**–**4** on a Varian VXR300 instrument. ^1^H and ^13^C NMR spectra were acquired at 300 K using deuterated chloroform (CDCl_3_) as solvent. Chemical shifts (d) are reported in parts per million (ppm) relative to the residual solvent peak as an internal reference and coupling constants (*J*) are reported in hertz (Hz). Spin multiplicity is reported as: s = singlet, br s = broad singlet, d = doublet, t = triplet, q = quartet, m = multiplet. Compounds were named following IUPAC rules, as applied by ChemBioDraw Ultra (version 14.0). Mass spectra and purity determinations were recorded on a Waters ZQ 4000, Waters XevoG2-XSQTof, acuity arc-QDA LC-MS apparatus with electrospray ionization (ESI) in positive mode. UPLC–MS analyses were run on a Waters ACQUITY UPLC/MS system, consisting of a QDa mass spectrometer equipped with an electrospray ionization interface and a 2489 UV/Vis detector (254 nm and 365 nm). The analyses were performed on an XBridge BEH C18 column (100 × 2.1 mm i.d., particle size 2.5 μm) with a XBridge BEH C18 VanGuard Cartridge precolumn (5 mm × 2.1 mm i.d., particle size 1.8 μm). The mobile phase was composed of H_2_O (0.1% formic acid) (A) and MeCN (0.1% formic acid) (B), using the following gradient: 0−0.78 min, 20% B; 0.78–2.87 min, 20−95% B; 2.87–3.54 min, 95% B; 3.54–3.65 min, 95–20% B; 3.65–5.73, 20% B. Flow rate: 0.8 mL/min. **1** showed a purity of ≥98%.

### 2.2. Synthesis of N-(2-(Benzyloxy)benzyl)ethanamine *(**2**)*

A solution of 2-(benzyloxy)benzaldehyde (0.5 g, 2.36 mmol) and ethylamine (0.127 g, 2.83 mmol) in toluene (10 mL) containing MgSO_4_ was stirred at room temperature for 1 h, then, the solvent was evaporated under vacuum. The resulting residue was rinsed with MeOH (35 mL), and NaBH_4_ (0.11 g, 2.83 mmol) was added and stirring continued for a further 48 h. After removing the solvent, the mixture was made cautiously acidic with 2N HCl and extracted with diethyl ether (3 × 15 mL). The aqueous phase was made basic with Na_2_CO_3_ and extracted with CH_2_Cl_2_ (3 × 25 mL). The organic phase was separated, dried, and evaporated to give 0.41 g (72% yield) of the title compound as a transparent oil. ^1^H-NMR (CDCl_3_, 300 MHz) δ 1.19 (t, *J* = 6.6 Hz, 3H), 2.78 (q, *J* = 6.3 Hz, 2H), 4.00 (s, 2H), 5.17 (s, 2H), 6.97–7.02 (m, 2H), 7.28–7.50 (m complex, 7H).

### 2.3. Synthesis of 2-(6-((2-(Benzyloxy)benzyl)(ethyl)amino)hexyl)isoindoline-1,3-dione *(**3**)*

To a solution of **2** (0.41 g, 1.70 mmol) in DMF (15 mL), *N*-(6-bromohexyl)phthalimide (0.264 g, 0.85 mmol) and KI (0.7 g, 5.11 mmol) were consecutively added, and the reaction mixture was stirred for 12 h at 90 °C. After cooling to room temperature, the mixture was acidified with aqueous KHSO_4_ and extracted with CH_2_Cl_2_ (3 × 25 mL). The organic phase was separated, dried, and evaporated to give a crude material, which was purified by flash chromatography. Eluting with CH_2_Cl_2_/petroleum ether/MeOH/aqueous 33% ammonia (4.5:5.0:0.5:0.05) afforded 0.2 g (50% yield) of the title compound as a yellow oil. ^1^H-NMR (CDCl_3_, 300 MHz) δ 1.06 (t, *J* = 6.9 Hz, 3H), 1.33 (s, 2H), 1.51 (s, 2H), 1.67 (t, *J* = 6.6 Hz, 2H), 2.49 (t, *J* = 6.7 Hz, 2H) 2.54–2.61 (m, 2H) 3.69 (t, *J* = 6.3 Hz, 4H), 3.96 (s, 2H), 5.11 (s, 2H), 6.97–7.05 (m, 3H), 7.34–7.49 (m complex, 6H), 7.70–7.73 (m, 2H), 7.84–7.87 (m, 2H).

### 2.4. Synthesis of N^1^-(2-(Benzyloxy)benzyl)-N^1^-ethylhexane-1,6-diamine *(**4**)*

A solution of **3** (0.2 g, 0.4 mmol) and NH_2_NH_2_•H_2_O (0.12 mL, 2.4 mmol) in EtOH (5 mL) was refluxed for 1 h. After cooling to room temperature, the resulting suspension was filtered, and the filtrate was evaporated under vacuum. The obtained crude mixture was dissolved in H_2_O acidified with KHSO_4_ and extracted with Et_2_O (3 × 20 mL). Then, the aqueous phase was made basic with NaHCO_3_ and extracted with CH_2_Cl_2_ (3 × 20 mL). The organic phase was separated, dried, and evaporated to give 0.12 g (88% yield) of the title compound as a yellow oil. ^1^H-NMR (CDCl_3_, 300 MHz) δ 1.06 (t, *J* = 6.8 Hz, 3H), 1.30–1.47 (m, 8 H), 2.45–2.71 (m. 6H), 3.69 (s, 2H), 5.11 (s, 2H) 6.92–7.51 (m complex, 9H).

### 2.5. Synthesis of 2-((6-((2-(Benzyloxy)benzyl)(ethyl)amino)hexyl)amino)-5-((6-(ethyl(2-(phenoxymethyl) benzyl)amino)hexyl)amino)cyclohexa-2,5-diene-1,4-dione *(**1**)*

A solution of **4** (0.12 g, 0.35 mmol) and 2,5-dimethoxy-1,4-benzoquinone (0.029 g, 0.17 mmol) in EtOH (6 mL) was stirred for 3 h at 60 °C. After cooling to room temperature, stirring was continued for a further 2 h. Then, the solvent was evaporated, and the resulting precipitate was washed with Et_2_O to give 0.12 g (85% yield) of the title compound as a red solid. ^1^H-NMR (CDCl_3_, 400 MHz) δ 1.04 (t, *J* = 7.1 Hz, 6H), 1.18–1.36 (m, 8H), 1.36–1.52 (m, 4H), 1.53–1.68 (m, 4H), 2.45 (t, *J* = 6.9 Hz, 4H), 2.55 (q, *J* = 7.0 Hz, 4H), 3.09 (q, *J* = 13.4, 6.8 Hz, 4H), 3.66 (s, 4H), 5.09 (s, 4H), 5.29 (s, 2H), 6.57 (t, *J* = 6.6 Hz, 2H), 6.86–7.01 (m, 4H), 7.19 (td, *J* = 7.7, 1.4 Hz, 2H), 7.29–7.35 (m, *J* = 7.2 Hz, 2H), 7.35–7.42 (m, 4H), 7.45 (d, *J* = 7.1 Hz, 4H); ^13^C-NMR (CDCl_3_, 100 MHz) δ 12.05, 26.98, 27.17, 28.34, 42.67, 47.85, 51.64, 53.41, 70.17, 92.75, 111.85, 120.77, 127.37, 127.87, 128.58, 130.43, 137.53, 151.41, 156.93, 178.20; MS (ESI^+^) *m*/*z* 785 [M + H]^+^. UPLC-MS purity (UV at 254 nm) 98%.

### 2.6. Physicochemical Descriptors

The physicochemical properties were calculated using the molecular descriptor calculator tool included in the QikProp module of the Canvas 3.2 platform [27,28]. The following descriptors were calculated: Molecular Weight (MW), ALogP, H-bond acceptors (HBA), H-bond donors (HBD), Topological Polar Surface Area (TPSA), and Rotable Bonds (RBs). For each property, a box plot was generated (Appendix A). 

### 2.7. Characterization of the Chemical-Space: Principal Component Analysis (PCA)

A PCA multivariate statistical method was followed to reduce to a 2D-visual representation the chemical space covered by the compound collection, as a function of the six molecular descriptors using the PCA module of the Canvas 3.2 platform [29]. The two drugs, verteporfin and dasatinib, were added to the screening panel as reference compounds. First, the exact number of principal components (PCs) able to explain the total variance (≥80%) of the original dataset was identified. Next, the correlation magnitude of all variables for defining each single PC was evaluated (loading value ≥±0.50), and PCA scores were calculated using the *k*-means clustering method as a function of six molecular descriptors. Consequently, PC score plot was generated [30,31]. Based on the distribution of PCA score values, the chemical space coverage was evaluated, and five structural complexity-based compound clusters were identified. Before performing the PCA, the original dataset was standardized by subtracting each value of the variable’s mean and dividing by its standard deviation.

### 2.8. Hierarchical Clustering Analysis 

A similarity-based clustering analysis was performed to identify putative compound groupings sharing the same chemical core structure using the hierarchical clustering tool in the Canvas 3.2 platform [29]. A single binary 2D-fingerprint was first calculated for each compound, considering an extended connectivity fingerprinting 4–ECFP4, in which the atoms and the bonds were distinguished by functional type and hybridization, respectively [30]. Next, a similarity–distance matrix was obtained based on the Tanimoto coefficient (=0.9092) [32], which was used for a bottom-up approach using the complete clustering-linkage as an agglomerative clustering method [33]. The same similarity matrix was also used as input-data for RStudio open-source software (https://rstudio.com/, accessed on 20 September 2021), to visually represent, as a dendrogram, the chemical similarities between molecules. With this aim, it was necessary to use the *hclust* algorithm, in order to translate the Canvas clustering matrix (*csv* file) to *tree* file format [34]. Sixteen compound clusters were obtained, four of which were single-populating clusters (singlets) [35]. The chemical molecules populating each single cluster were analyzed one by one and the proper common core was defined. Finally, the tree file was used as input for the iTOL online server (http://itol.embl.de/, accessed on 20 November 2021) for displaying the clustering treemap [36,37].

### 2.9. Cell Culture

HaCat cells were maintained in DMEM medium (Dulbecco’s modified Eagle’s medium; composition: 100 mg/L Glucose, Sodium Bicarbonate and pyridoxine) supplemented with 2 mM L-Glutamine and 10% FBS (fetal bovine serum). A2780 and A2780/CP cell lines were maintained in RPMI medium, containing 10% FBS. HT29, HCT116, and HCA46 (colorectal cancer cell lines) were maintained in DMEM medium. Cells were incubated at 37 °C in a humidified atmosphere containing 5% CO_2_. All studies were performed in Mycoplasma-negative cells. All cell lines were obtained from ATCC and the European collection of cell cultures, ECACC via Sigma [38]. Verteporfin was purchased from Sigma Aldrich, diluted in DMSO and the aliquots were stored at −20 °C. Deslanoside and Digoxin were purchased from Sigma Aldrich. All treatments, including verteporfin were performed in dark conditions, to avoid any issue due to the photosensitivity of this molecule [39]. 

### 2.10. Cell Viability Assay

A growth inhibition assay was performed to assess cell viability. Cells were seeded in 24-well plates at a density of 5 × 10^5^ cells per well. After 24 h, cells were exposed to increasing concentrations of the compounds for 18 and 48 h. 3-(4,5-dimethylthiazol-2-yl)-2,5-diphenyltetrazolium bromide reagent (MTT) (Sigma Aldrich) was added to the cells at a final concentration of 0.5 mg/mL, and then the cells were incubated for 4 h at 37 °C in a humidified incubator with 5% CO_2_, until purple precipitates were visible. The medium was removed, and the cells were washed once with PBS. To dissolve the formazan precipitates, DMSO was added to each well and the absorbance was read at OD = 535 nm. The drug concentration resulting in 50% inhibition of cell growth (EC_50_) was calculated. 

### 2.11. Dual Luciferase Reporter Assay

Twenty-four hours before the transfection, HaCat cells were seeded into 24-well plate (5 × 10^5^ cells/well). The day after, the cells were transiently co-transfected with pGL4 Firefly luciferase ‘experimental’ vector (125 ng) (Promega), in which was inserted a YAP/TEAD response element (YRE; 8XGT-IIC lux) at the level of the upstream promoter of luciferase gene; pGL4 Renilla luciferase ‘control’ vector (10 ng) (Promega) and YAP expressing plasmid ‘activator’ plasmid (42 ng), using Lipofectamine 2000 Reagent (Invitrogen). After twenty-four hours from the transfection, the cells were treated with the compounds for 18 h. The concentrations used for the treatment were the IC_50_ for each compound derived from the cell viability assay at 48 h. Verteporfin was used as a reference compound. The dual luciferase reporter assay was performed according to the manufacturer’s instructions (Promega).

### 2.12. Immunoblotting

Cells were washed twice in ice-cold phosphate-buffered saline (PBS) and then lysed in RIPA Lysis Buffer (1% sodium deoxycholate, 20 mM Tris-HCl (pH 7.6), 150 mM NaCl, 1% NP-40, 1 mM EGTA, 1 mM Na_2_EDTA, 1 mM Na_3_VO_4_, Na_2_MoO_4_, 1 mM PMSF and protease inhibitor cocktail) (Sigma-Aldrich Corporation). The resulting suspension was centrifuged for 30 min, 14,000 rpm at 4 °C. The supernatant was collected, and the protein concentration was determined using Bradford Reagent (Sigma Aldrich). Cell lysates were incubated with 4× SDS sample buffer for 5 min at 100 °C and then were run on SDS-PAGE gels and transferred to PVDF membranes. Western blot analysis was performed using anti-YAP (clone 4D11.1, Millipore, dilution 1:1000), anti phospho-YAP Ser127 (clone D9W2I, Cell signaling, dilution 1:1000), and anti-Actin (clone C4, Millipore, dilution 1:2500). Protein bands were detected by enhanced chemiluminescence (ECL, GE Healthcare). Band density was calculated by using Image J software.

### 2.13. Immunofluorescence Staining

HT29 cells were seeded in multi 24-well plates (5 × 10^5^ cells/well) and the day after were treated with the compounds for 24 h. Cells were fixed for 10 min in 4% paraformaldehyde (PFA). The fixed cells were permeabilized with 0.3% Triton X-100 in PBS for 2 min on ice. After blocking in 3% BSA for 30 min, slides were incubated with the first antibody diluted in 1% BSA for 1.5 h. After washing with PBS, the slides were incubated with Alexa Fluor 568-conjugated secondary antibodies (1:1000 dilution) for 1.5 h. Nuclei were stained with DAPI for 3 min. Then, the slides were washed and mounted with antifade reagent (Vectashield, Vector Laboratories-H1000).

### 2.14. Real Time PCR (qPCR)

Total RNA was extracted using Triazol Reagent and a PureLink RNA Mini Kit, the miRNeasy Mini Kit (Invitrogen), according to the manufacturer’s instructions. Reverse transcription was performed with 0.5 μg of total RNA using SuperScript first-strand, synthesis for RT-PCR (Invitrogen). The cDNA strand generated was used for PCR amplification in a total volume of 20 μL. Real-time PCR was performed with 100 ng of cDNA using the Power SYBR green PCR master mix (Applied Biosystems). Samples were amplified in a CFX96 Touch™ Real-Time PCR Detection System. Each sample was run independently in triplicate. The amplified were analyzed using Biorad CFX software. For each sample, three replicates were acquired, and Student’s t test was performed to evaluate statistical significance. The amount of target genes expressed was normalized using r18s.

### 2.15. Statistical Analysis

All data were obtained using the mean of three experiments. Statistical analysis was performed by using an unpaired *t*-test. A *p*-value of <0.05 was considered significant. Analyses were performed using GraphPad Prism v7 (La Jolla, CA, USA).

## 3. Results

### 3.1. Chemical Library Collection

With the goal of identifying YAP-TEAD inhibitors in a timely and cost-effective manner, and tailoring them for further development, we assembled a small chemical library of 27 entries, selected from a collection of about 2000 compounds within the databases of our academic laboratories. We included a few commercial compounds, such as pristimerin, digoxin, deslanoside, and verteporfin with known connection with the Hippo pathway. The chemical structures of the included compounds are reported in Appendix A, whereas the main scaffolds are depicted in Appendix A. The library size was restricted to lead-like compounds based on: (i) chemical diversity, (ii) potential connection established by the compounds’ target with YAP-TEAD and the Hippo pathway activity, and (iii) structural features suitable for PPI inhibition. This allowed us to provide a reasonable coverage of chemical space (see Section 3.2), and high-quality starting points (available information on synthesis, physico-chemical properties and bioassay data) for further optimization studies. To develop a more tailored selection and to examine the connection to YAP–TEAD and Hippo pathway activity (point ii), we performed a metadata analysis, reported in detail in the Appendix A (pages 3, 4), in which the biological activity of all compounds was described. Then we cross-examined the networks of PPI (interactome) of the biochemical and functional distances between the core proteins of the Hippo pathway (TEAD1–4, LATS1/2, YAP1, MOB1A, MOB1B, STK4, SAV1), representing the neighbor upstream kinome of YAP (Figure 1), and the compounds’ target. With this aim, we adopted suitable bioinformatic tools such as STRING www.string-db.org, accessed on 25 January 2022) and PANTHER (www.pantherdb.org, accessed on 25 January 2022). This work was preliminary to the compound selection and is reported in detail in Appendix A. On the basis of the predicted interactome generated with the Hippo pathway components, it was possible to obtain three groups from the selected library, helping in a further reduction of the library.

Our library includes registered drugs (~7%) and approximately 93% preclinical compounds coming from in-house molecular design expertise and synthesis efforts in various drug discovery projects (Appendix A). Particular attention was given to the presence of hydrophobic, rigid, and aromatically-privileged scaffolds, able, in principle, to modulate PPI [40,41]. The chemical library collected for the cell-based screening assay included different chemotypes having a common scaffold close to molecules known to be able to act either directly or indirectly on the terminal effectors YAP/TAZ and TEAD of the Hippo signaling pathway. 

All compounds had never been studied in the context of the Hippo pathway. The compounds selected for the library carry structural features such as those reported in the iPPI-DB database containing protein–protein interactions modulators (https://ippidb.pasteur.fr, accessed on 25 January 2022) [27].

### 3.2. Chemoinformatic Analysis: Physicochemical Descriptors and PCA

A chemoinformatic in silico analysis was performed to characterize the physical and chemical properties of the 27-compound collection using the QikProp module of the Canvas 3.2 platform. Verteporfin and dasatinib were added to the library as reference compounds with a known biological profile in YAP–TEAD activity modulation. For each molecule, the calculated descriptors Molecular Weight (MW), AlogP, number of H-bond acceptors (HBA) and H-bond donors (HBD), Topological Polar Surface Area (TPSA), and number of Rotatable Bonds (RBs) were analyzed (Appendix A). The MW of the compounds ranged from 206 g/mol to 1344 g/mol. The calculated lipophilicity (AlogP) ranged from −1.18 to 9.05. The number of HBA was 0–19; whereas the number of HBD varied from 0 to 9. Extending the RO5 evaluation to include properties associated with favorable bioavailability, the library showed a TPSA in the range 24–282 Å^2^, with between 1 and 28 rotatable bonds, with a mean of 8.4. Appendix A shows the distribution of the parameters in the library, and assuming no more than one violation of the rule, 51.7% of the entire library resulted as in accordance with Lipinski’s ‘rule of five (RO5)’ [42]. Therefore, the library contains compounds that show drug-like properties; thus, being useful for further studies in the hit-to-lead phase.

Principal component analysis (PCA) was performed for evaluating the chemical space coverage of our molecule collection as a function of the six physicochemical properties previously defined. In PCA analysis, each principal component (PC) is a linear combination of all original variables involved, and PCs are defined in such a way that the variance of the scores is maximal. Based on the screen-plot in Appendix A, reporting the total amount of variance explained by each single PC (eigenvalue), it was possible to show that the first two PCs cumulatively retrieved 87.2% of the variance. By setting 80% as the threshold of total variance, they are, hence, suitable for giving a comprehensive representation of the relationships between the variables of the entire original dataset [43,44]. Considering the loading value of each single variable [44], it was possible to show that the MW and TPSA variables had the highest contribution to defining PC1 (significant correlation ≥ ±0.50), while AlogP had a negligible role in explaining the variation due to PC1; in the same way, PC2 resulted as strongly correlated with AlogP and RBs, whereas the remaining variables had a lesser relevance for assessing the variation due to this principal component. Appendix A summarizes the loading values for all six original properties. In Figure 3, the PCA scores of the entire chemical collection are shown as a score-plot, displaying a 2D-visual representation of the chemical space with identification of compound clusters based on complexity structure similarities, alongside the variable correlations. In our specific case, the PCA results were mapped using a traffic-light system, from blue (low complexity) to bright red (high complexity), with moderately complex molecules in violet-purple. Based on the results, it was possible to determine that the original database consisted of more blue-to-purple chemical entries; therefore, revealing a medium-high level of molecular complexity for our selected compound library. 

Based on the map-distance, our library spans different regions of the chemical space in the 2D reference frame. In this graph, single dots located at opposing region of the score plot correspond to chemical compounds being strongly dissimilar and occupying different and far away regions of the chemical space. Based on the distribution of PCA scores, it was possible to identify five main compound clusters, occupying similar regions of the chemical space. The five identified clusters are the following: cluster 1 (blue): antipao, **ULA26**; cluster 2 (persian blue): **UCM871**, **ST11cis**, **ST11trans**, **UPR1260**, **IA5**, **ACG35**, **JCM24**; cluster 3 (navy blue): **EU11**, pristimerin, **UPR1320**, **URB694**, **URB913**; cluster 4 (violet-purple): memoquin, **ADC3**, **RS34**, **PCM129**, **UPR1355**, **BAT33**, **UPR1268**, dasatinib, **DA15**, **SVT118**, verteporfin; cluster 5 (magenta-cherry-red): digoxin, **CDM38**, deslanoside. 

In addition, aiming to identify putative core structures shared by the molecules populating the entire collection, a similarity-based hierarchical-clustering analysis was performed. Sixteen compound clusters were first identified based on functional type and hybridization grade of atoms and bonds, respectively. Among these, four clusters were singlet clusters, while the molecules populating the remaining 12 clusters were further visually inspected, and compound groupings sharing the same chemical core-structure were grouped together: i—3-vinylpyridine; ii—1,3-dimethoxybenzene; iii—2,5-diamino-cyclohexa-2,5-diene-1,4-dione; iv—N-(2-((3-methoxy-phenyl)amino)ethyl)acetamide; v—phenol; vi—chlorobenzene; vii—6,7-dimethoxyquinazoline-2,4-diamine; viii—acridin-9-amine; ix—1H-pyrrol-1-amine; x—1H-indole; xi—5α,14α-androstane; xii—methylcyclohexa-2,4-dienecarboxylate (Figure 4). We also performed a similar analysis using the DenovoProfiling webserver (http://denovoprofiling.xielab.net, accessed on 20 November 2021), which provides structural identification and chemical space exploration relative to compounds in the public database, i.e., PubChem, and the collected results were very similar (data not shown). The compound collection was then analyzed in details in the following sections [45,46,47,48,49,50,51,52,53,54,55,56,57,58,59,60,61,62,63,64,65,66].

### 3.3. Chemistry

Most of members of the in-house library were not resynthesized, and their synthetic and characterization data are reported in their respective publications (quinones: memoquin [45], **SVT118** [46], **IA5** [48], **EU11** [66]; quinazolines and quinolines (**CDM38**, **BAT33**, **UPR1268**) [49,50,51]; stilbenes (ST11 cis/trans) [52]; thiazolidindione (**ACG35**) [53]; diketopiperazine (**JCM24**) [54]; tetrahydroacridine (**RS34**) [55]; dibenzodiazepinone (**ADC3**); carbazole (**ULA26**) [56]; steroids (pristimerin, **PCM129**, **UPR1355**, digoxin, deslanoside) [57,60,61,62]; carbamates (**URB694**, **URB913**) [58,59]; anilino-alkyl-amides (**UCM765**, **UCM871**) [63,64] (Figure 2). One of the selected compounds (**DA15**) is unpublished, and we report the synthetic procedure below. As depicted in Figure 1, amine **2** was obtained through the reductive amination of 2-(benzyloxy)benzaldehyde with ethylamine. Then, the nucleophilic substitution reaction between **2** and *N*-(6-bromohexyl)phthalimide afforded compound **3**. Removal of the phthalimide protecting group was performed, with hydrazine leading to the deprotected primary amine **4**, which was then reacted with 2,5-dimethoxy-1,4-benzoquinone, providing **1** (**DA15**) in a good yield (85%). **1′** identity was confirmed by analytical (HPLC), full scan MS (ESI+), and spectroscopic data (NMR, see reported spectra in Appendix A). 

### 3.4. Identification of Compounds with the Potential to Inhibit Cell Growth

To identify efficient cell growth inhibitors, we screened the chemical library of 27 compounds (Appendix A) with the Human Keratinocyte cell line (HaCaT). The HaCaT cell line has a high capacity to proliferate and differentiate in vitro and is widely used for Hippo pathway studies. HaCaT cells were treated with all compounds at several concentrations (0.1–20 µM) for 18 and 48 h (Appendix A, respectively), and the corresponding IC_50_ values were measured by performing an MTT assay (Appendix A). To select active compounds (Table 1), we evaluated the IC_50_ at 48 h, while we considered the IC_50_ at 18 h, to ensure a high percentage of viable cells, necessary for the YAP–TEAD complex activity study, described below. In Figure 5, we show the growth inhibition effects of the compounds that at 10 µM reduced cell viability, within a range of IC_50_ between 0.2 and 7.3 µM at 48 h (Table 2). Three of these compounds, digoxin, deslanoside, and pristimerin, showed activity at nanomolar concentrations (IC_50_ values: 0.20, 0.25 and 0.80 µM, respectively, Table 2), and the other four compounds (**IA5**, **DA15**, memoquin, and **UPR1268**), inhibited cell growth at concentrations lower than 8 µM. 

The most active compounds were progressed to further studies (Figure 2). Among the seven active compounds, three quinones [45,66], one quinoline [51], and three steroid derivatives [57,62] were included. Remarkably, quinones, such as memoquin, **DA15**, **IA5**, or quinolines, such as **UPR1268**, were pinpointed as inhibiting PPI, by abolishing pivotal π-π or π-cation interactions, deemed to be the hot spots for protein–protein interface formation [40], or by bearing multiple aromatic rings and conformational stiffnesses [41]. Quinone, bearing a polyamine backbone (i.e., memoquin or **DA15**), has an additional chance of recognizing multiple anionic sites of the protein–protein interface by assuming different protonated conformations.

Chemical entities belonging to the steroid class, such as the triterpenoid pristimerin [57] has demonstrated antiproliferative properties. Pristimerin is known to inhibit cell invasion, migration, and proliferation in colon cancer cells [67], while **UPR1268** is effective against malignancy transformation and progression, through pathways inferred from the literature [65] in gefitinib-resistant non-small-cell lung cancer [68]. Cardiac glycosides (such as digitoxin [24]) can promote the nuclear translocation of YAP in YAP low-level oncotypes [69], and decrease the relative ratio of pYAP to total YAP in cancer cell lines [24]. As previously suggested in a computational model by Sudol et al. [23], the binding site of this class of molecules was predicted to be at the WW1 domain level, with subsequent displacing of cytoplasmic inactivators [13]; however, the mechanism is still unclear. Based on this information, the selected compounds showed potential for more specific YAP–TEAD modulation studies.

### 3.5. Inhibition of the Activity of the YAP–TEAD Complex 

We next tested the compounds’ ability to specifically modulate YAP–TEAD activity. We performed a specific gene-reporter assay based on Luciferase signal variation detection [69]. By cloning a YAP responsive element upstream, the promoter sequence of Luciferase reporter gene, it was possible to evaluate the ability of the selected YAP-target compound library, to modulate gene transcription [70]. First, we co-expressed pGL4 Firefly luciferase, pGL4 Renilla luciferase, and YAP plasmids in HaCaT cells, widely used for Hippo pathway studies, and after 24 h we treated the cells with **IA5**, **DA15**, memoquin, **UPR1268**, pristimerin, digoxin, and deslanoside for 18 h, at their IC_50_ value calculated at 48 h. VP, known to disrupt YAP–TEAD complex at 18 h in uveal melanoma cells, was used as a reference compound in all experiments [16,71,72]. We found a reduction of the luciferase activity, showing a significant decrease of the YAP–TEAD complex activity after the treatment with the selected compounds, comparable with the reference molecule, VP (Figure 6). These results may be an important step in studying how these compounds would affect the YAP-TEAD activity in cancer cells.

### 3.6. Cancer Cell Growth Inhibition and YAP Protein Levels and Phosphorylation Changes

Much evidence highlights that YAP-TEAD activity is essential for cancer initiation and the growth of most solid tumors [1,7]. To assess the inhibitory effect of our compounds on cancer cells, we selected three colon cancer cell lines (HCA46, HCT116, and HT29), and two ovarian cancer cell lines (A2780 and A2780/CP). It has been reported that cell proliferation is affected by YAP knockdown [3,73], and, therefore, the experiments proposed had the rationale of addressing the potential capacity of the tested compounds to modulate YAP-TEAD activity. Cancer cells were treated for 48 h with various concentrations of the compounds, followed by MTT assay. Most of the compounds inhibited the cell growth, as shown in Table 2. Digoxin, deslanoside, and pristimerin showed IC_50_ values at nanomolar concentrations. **IA5**, **DA15**, memoquin, and **UPR1268** inhibited cell growth in all cancer cell lines, showing an IC_50_ value between 1 and 14 µM. 

To study the effects of our compounds in colorectal cancer cells, in which YAP plays a crucial role in the aggressiveness and metastasis [74], we first identified the protein expression profiles of YAP and its phosphorylated form at Ser127 residue, in a panel of colon cancer cell lines (SW116 and HCA46 sensitive to Fluorouracil (5FU), LoVo, Caco2, HT29 and HCT116 less sensitive to 5FU) (Figure 7A). Western blotting analysis showed that YAP protein levels were differently expressed in these cell lines, as well as their phosphorylation (Figure 7A). To evaluate whether our molecules might affect the YAP and pYAP-Ser127 protein level, HT29 cells were treated with the selected compounds for 24 h at their IC_50_ values. The compounds **IA5**, **UPR1268**, digoxin, and deslanoside reduced the total YAP and pYAP-Ser127 protein levels, **DA15** increased the total YAP protein levels but not its phosphorylated form (pYAP-Ser127), and memoquin enhanced both the total YAP protein and pYAP-Ser127 (Figure 7B,C). Since YAP phosphorylation at Ser127 residue is an important indicator of YAP function activity and its subcellular localization, the ratio between pYAP-Ser127 and total YAP was quantified as well (Figure 7D). Notably, we found that **IA5** increased the levels of pYAP-Ser127/YAP ratio, indicating that this compound might promote the cytoplasmic YAP retention, with a consequent reduction of nuclear YAP translocation and its activity. All other compounds, such as the reference compound verteporfin, showing a reduction of YAP-TEAD activity in luciferase assay, unexpectedly exhibited a reduction of pYAP-Ser127/YAP ratio (Figure 7D). This prompted us to hypothesize that they might modulate the protein YAP amount/activity with a different mechanism of action with respect to **IA5** compound. 

Furthermore, we confirmed the effect of **IA5** on the YAP protein levels, by using immunofluorescence staining in HT29 cells. The HT29 cells were treated with **IA5** and all other compounds for 24 h (Figure 7E). The results showed a significant decrease of the YAP protein level, both in the nucleus and in the cytoplasm, compared with the verteporfin treatment. Digoxin and deslanoside, as well as digitoxin in a previous study [69], promoted YAP nuclear retention, agreeing with the attenuated YAP-Ser127 phosphorylation level that we observed (Figure 7B–D). All other compounds did not display any significant variation of the YAP level with respect to control cells.

### 3.7. IA5 Reduces YAP-TEAD Target Genes Expression

Finally, to understand the mechanism by which our compounds inhibited the cell growth, we examined the expression levels of YAP mRNA and YAP-TEAD target gene expression in two colorectal cancer cell lines, HCT116 and HT29. The mRNA levels of YAP and of the two specific target genes, cysteine-rich angiogenic inducer 61 (CYR61) and connective tissue growth factor (CTGF), were evaluated using quantitative PCR (qPCR), in cells treated for 48 h (Figure 8). We found that the **IA5**, similarly to VP, induced a significant decrease of YAP mRNA level and its target genes, CYR61 and CTGF, in both HCT116 and HT29 cell lines, affecting the YAP-TEAD complex activity. These data are in accordance with the luciferase assay results (Figure 7B) and the phosphorylation variation of YAP protein (Figure 7B–D), suggesting that the **IA5** compound might represent a potential candidate for cancer cell growth inhibition, by targeting the YAP-TEAD complex activity. Digoxin and deslanoside decreased the protein expression of YAP (Figure 7C) and increased CTGF and Cyr61 mRNA levels (Figure 8A). It is reasonable that in our experimental conditions, TAZ protein may have replaced YAP after its reduction. Indeed, YAP and TAZ are generally thought to be functionally redundant [75]. All other compounds did not show any significant reduction in the YAP mRNA levels and CYR61, CTGF target genes.

## 4. Discussion 

Since the YAP–TEAD complex is an interesting target in cancer, modulating the interaction of YAP/TAZ with transcriptional factor, TEAD, may be an effective strategy for cancer therapy [11]. Currently there are no clinically available drugs targeting the YAP/TAZ–TEAD interaction for cancer treatment. For this reason, following a repurposing strategy, in this study we collected and screened a library of 27 small molecules, supposed to modulate TEAD gene expression. In all experiments, we also included VP, the first small molecule reported as a YAP-TEAD disrupter [17,76]. First, a chemoinformatic in silico analysis was performed to characterize the physical and chemical properties of the 27-compound collection, using the Chemioinformatic platform. We could conclude that our compounds were characterized by a medium-high level of molecular complexity. The library was characterized by high molecular diversity, in which different clusters were identified. Then we evaluated the effect of these compounds on the cell viability in HaCaT cells, a model widely used for Hippo pathway studies, and measured the IC_50_ by MTT assay. Among these compounds, we selected seven out of 27, as able to inhibit the cell growth at low concentration. Next, we tested their capability to reduce the YAP–TEAD complex activity in cells, using an luciferase assay. We found that the tested compounds reduced the YAP-TEAD activity with respect to the control cells. It is well known that YAP plays an important role in regulating cell tumor proliferation [7], and its overexpression is associated with poor prognosis in cancer patients [4]. Consequently, we chose to evaluate the effect of the selected compounds on the YAP protein level, and YAP phosphorylation status in three colorectal cancer cell lines and two ovarian cancer cell lines. We found that the quinone derivative **IA5** reduced the total YAP protein levels and increased the pYAP-Ser127/YAP ratio in HT29 cells, with respect to controls, indicating that **IA5** might inhibit the nuclear YAP translocation, followed by a reduction of YAP-TEAD activity. This might be an important feature for a potential compound capable of targeting the YAP–TEAD complex, inhibiting the cancer proliferation. Moreover, we found that digoxin and deslanoside reduced the total protein YAP, but also the pYAP-Ser127/YAP ratio, suggesting that they might operate similarly to digitoxin, inducing the YAP nuclear retention and enhancing YAP activity [69].

Furthermore, we confirmed the impaired YAP protein expression, and YAP intracellular localization, in HT29 cells using an immunofluorescence technique. The HT29 cells, after treatment with **IA5**, digoxin and deslanoside for 24 h, displayed a drastic reduction of YAP protein when cells were treated with **IA5** with respect to control cells, while the two digitoxin derivative molecules (digoxin and deslanoside) increased the YAP protein translocation into the nucleus. Notably, YAP interacts with TEAD in the nucleus and induces expression of genes involved in the regulation of cell growth, proliferation, and survival, including connective tissue growth factor CTGF and cysteine-rich angiogenic inducer 61, CYR61 [77]. Then, we examined the expression of these two YAP-TEAD target genes in two colorectal cancer cell lines (HCT116 and HT29), and, as expected, we observed that **IA5** reduced the mRNA level of CYR61 and CTGF1, with respect to control cells. The reduction was comparable with the reference molecule, VP. This result may suggest **IA5** as a potential inhibitor of TEAD activity. Quinone moieties are widely distributed in nature and are the key pharmacophore referred to the 2-(4-hydroxyanilino)-naphthoquinone scaffold (quinone in Figure 2) of many clinically important anti-cancer drugs, such as anthracycline, daunorubicin, doxorubicin, mitomycin, mitoxantrones, and saintopin. Quinones are also known as highly redox active molecules, leading to the formation of reactive oxygen species (ROS), which can alter the redox balance within cells through the formation of oxidized cellular macromolecules, including lipids, proteins, and DNA; accounting at least in part, for the reduced viability of all our cell lines observed with the compounds, **IA-5**, digoxin, and deslanoside [78,79]. In addition, to reinforce this effect, it was also shown that inactivation of YAP decreases antioxidant gene expression, leading to enhanced oxidative stress-induced cell death through downregulation of catalase and MnSOD [80]. 

The putative interference behind the YAP-TEAD activity inhibition by **IA5** might result in the modulation of PPI, ultimately altering the TEAD’s CYR61, CTGF gene expression. In particular, the 2-(4-hydroxyanilino)-naphthoquinone scaffold of **IA5** may establish hydrogen bonds, aromatic contacts, and a favorable interaction between the central electron-poor quinone ring and the electron-rich peptidic carbonyls aromatic residues; all recognized as common key motifs in iPPI [81]. While **IA5** has been reported for its capability to affect the cell cycle [66] and for its selective antiproliferative effect on cancer cell lines [82], to the best of our knowledge this is the first time that its activity was evaluated in the context of the Hippo signaling cascade. The detailed mechanism of action of **IA5** compound will be explored in future experiments, to guide further medicinal chemistry work, leading to more selective YAP-TEAD activity modulators.

## Data Availability

Data supporting the reported results can be found at the following address: https://zenodo.org/deposit/4624361, accessed on 20 November 2021. Raw data file associated to cellular studies of the manuscript are reported in the file “raw data_YapExp_cellular experiments.xlsx” and deposited in the following website https://zenodo.org/deposit/4624361, accessed on 20 November 2021. Raw data file associated with Figure 7 of the manuscript is reported in the following file “RAW IMAGEWB_Figure7.pdf” and deposited in the following website https://zenodo.org/deposit/4624361, accessed on 20 November 2021.

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
