# Peer review of "Identification of a Quinone Derivative as a YAP/TEAD Activity Modulator from a Repurposing Library"

_pharmaceutics, 2022, doi:10.3390/pharmaceutics14020391_

Round 1
Reviewer 1 Report
The present article reports the discovery of a quinone derivative through a not-conventional procedure which started from a (very/too) small library of 27 compounds. This library was in silico analyzed for characterization of physicochemical properties. Then the compounds were screened in a cell viability assay on HaCaT cells with verteporfin as reference and seven compounds were selected. The three steroids have submicromolar IC50’s while the other compounds have molar to decimolar IC50’s. Then the effects of these compounds were evaluated on the YAP-TEAD activity in the classical luciferase assay at their IC50 previously measured and then on HT29 cells which expressed modestly YAP to assess the effect of the tested compounds on the phosphorylation of YAP. As expected VP, steroids, IA-5 and UPR1268 totally extinguished YAP and pYAP while memoquin overexpressed YAP and pYAP and DA-15 overexpressed pYAP and down-regulated YAP. Finally the effect of the selected drugs and VP on the expression of the target genes Cyr61 and CTGF were measured on HT29 and HCT116 cell lines.
Many points have to be clarified before possible publication of this work, including the following ones:
- Using VP as reference needs to perfectly control the light intensity to avoid the risk of phototoxicity.
- The authors started from a very short set of compounds to identify one compound acting on the Hippo pathway without any idea of the mode of action
- Fig 8A suggested that HCT116 cells would be more convenient than HT29 cells to measure the effects of the drugs on YAP phosphorylation.
- Memoquin increased the protein expression of YAP while it has no effect on YAP mRNA level.
- Digoxin and deslanoside decreased the protein expression of YAP and increased CTGF and Cyr61 mRNA levels.
- Finally the quinonic function could simply explain the cytotoxicity of IA-5 by production of ROS as in the case of digoxin and deslanoside.
Author Response
Response to Reviewer 1
The present article reports the discovery of a quinone derivative through a not-conventional procedure which started from a (very/too) small library of 27 compounds. This library was in silico analyzed for characterization of physicochemical properties. Then the compounds were screened in a cell viability assay on HaCaT cells with verteporfin as reference and seven compounds were selected. The three steroids have submicromolar IC50’s while the other compounds have molar to decimolar IC50’s. Then the effects of these compounds were evaluated on the YAP-TEAD activity in the classical luciferase assay at their IC50 previously measured and then on HT29 cells which expressed modestly YAP to assess the effect of the tested compounds on the phosphorylation of YAP. As expected VP, steroids, IA-5 and UPR1268 totally extinguished YAP and pYAP while memoquin overexpressed YAP and pYAP and DA-15 overexpressed pYAP and down-regulated YAP. Finally the effect of the selected drugs and VP on the expression of the target genes Cyr61 and CTGF were measured on HT29 and HCT116 cell lines.
Many points have to be clarified before possible publication of this work, including the following ones:
Point 1: Using VP as reference needs to perfectly control the light intensity to avoid the risk of phototoxicity.
Response 1: we thank the Reviewer for notifying us. We are aware that using VP needs to perfectly control the light intensity to avoid the risk of phototoxicity. Based on the above reason, all VP treatments have been performed in dark conditions. We agree with the Reviewer's suggestion, and a specific sentence and Reference have been included in the Main Text (Material and Methods, section 2.9, page 7, lines 435-437).
Point 2: The authors started from a very short set of compounds to identify one compound acting on the Hippo pathway without any idea of the mode of action.
Response 2: We thank the reviewer for pointing this out and we apologize for not being clear. We clarified how the compounds have been selected in order to reduce library size suitably to our academic setting (page 7, section 3.1, lines 530-553). We have included more explanation in the main text. In summary, other than the already included criteria, we have also performed a metadata analysis on the known mechanism of action of the compounds (reported in the Supplementary Materials page 2 and 3, and Table S1).
To further understand the biological aspect of the selected compounds, we have performed a protein network enrichment analysis, based on freely available bioinformatic tools, such as STRING and PANTHER, starting from some selected core proteins and known targeted proteins for each compound reported in table S1. The description of the protein network analysis allowed us to recognize that some compounds may influence the Hippo kinase core proteins. The analysis results are reported in the Supplementary material 2 file associated with the manuscript. We added the following sentence in the main text: sub-heading Chemistry 3.1 pag 8, lines 545-553.
“By starting from the analysis of in-house libraries in comparison with known YAP-TEAD activity modulators, we restricted the library size to lead-like compounds based on: (i) chemical diversity, (ii) potential connection to YAP-TEAD and Hippo pathway activity, and (iii) features suitable for PPI inhibition. This allowed us providing a reasonable coverage of chemical space (see 3.2), and high-quality starting points (available information on synthesis, physico-chemical properties and bioassay data) for further optimization studies. To improve a more tailored selection and to examine the potential connection to YAP-TEAD and Hippo pathway activity (point ìì) we performed a metadata analysis (Supplementary Material pag 3,4), in which the biological activity of all compounds was described. Then we cross-examined the networks of protein-protein interactions to highlight the biochemical and functional distances between the core proteins of the Hippo pathway (TEAD1-4, LATS1/2, YAP1, MOB1A, MOB1B, SAV1) representing the neighbour upstream kinome of YAP (Figure 1) and the compounds’ target using suitable bioinformatic tools such as: STRING and PANTHER. This work was preliminary to the compound selection and is reported in details in the Supplementary Material 2.
--------------------------------------------------------------------------------
Point 3: Fig 8A suggested that HCT116 cells would be more convenient than HT29 cells to measure the effects of the drugs on YAP phosphorylation.
Response 3: we thank the Reviewer for the comment/suggestion. We have decided to use the HT29 cell line instead of HCT116 cell line based on the blot of the Figure 8A (in the revised Manuscript replaced with Figure 7B) showing that HCT116 cells have high steady-state level of YAP and pYAP, and as such, potentially less suitable to be used in our experiments. In fact, the risk of saturation of the bands may lead to potential low experiments reproducibility issues in the event that an increase in levels had occurred after treatments (….”Saturation of strong bands obscures actual differences between samples.”….., Lakshmi Pillai-Kastoori et al, Analytical Biochemistry, 2020, https://doi.org/10.1016/j.ab.2020.113608). In our case, after having observed HCT116 blot shown in Figure 8A, (now Figure 7A), with such high level of Yap proteins expression, we were expecting that the treatments could increase even more the levels of YAP and pYAP. To prevent potential signal detection problems we selected HT29 that showed a lower Yap proteins level. MEMOQUIN and DA15 showed an increase of the signal and we were able to detect the changes without the saturation issues.
Another criteria adopted to select HT29 and not HCT116 is that we have chosen to evaluate the cell line with similar expression levels of pYAP and YAP in the steady-state condition. Therefore, our choice of HT29 was more based on technical aspects of experiment reproducibility.
Point 4: Memoquin increased the protein expression of YAP while it has no effect on YAP mRNA level.
Response 4: we thank the Reviewer for this comment. We are aware that the result of quantitative PCR (qPCR) does not match the western blot result, but the end point of a biochemical determination is the expression of the related protein. In fact, the correlation between the expression levels of mRNA and protein are notoriously poor. The discrepancy is typically attributed to other levels of regulation between transcript and protein product (de Sousa Abreu, R, et al, Mol. Biosyst., 2009, doi:10.1039/b908315d; Vogel, C. & Marcotte, E. M., Nat. Rev. Gen., 2012, https://doi.org/10.1038/nrg3185; Maier, T., Güell, M. & Serrano, L., FEBS Lett., 2009, https://doi.org/10.1016/j.febslet.2009.10.036; Koussounadis et al., Scientific Reports, 2015, DOi: 10.1038/srep). Furthermore, there are several well-known post-translational modifications (PTMs) of YAP that affect its protein stability, transcriptional activity and subcellular localization (Fangjie Yan et al, BBA - General Subjects, 2020, https://doi.org/10.1016/j.bbagen.2019.07.006).
A specific post-translational modification of YAP may occur after MEMOQUIN treatment, enhancing its stability.
Point 5: Digoxin and deslanoside decreased the protein expression of YAP and increased CTGF and Cyr61 mRNA levels.
Response 5: we thank the Reviewer for notifying us. We are aware that the result of the immunoblot (decreased expression of YAP protein) does not match the quantitative PCR (increased CTGF and Cyr61 mRNA levels). We agree with the Reviewer's comment. There are different possibilities to explain this behavior, as already observed in the case of memoquine. In this case, our hypothesis is that in our experimental conditions, TAZ protein may replace YAP after its reduction. Indeed, YAP and TAZ are generally thought to be functionally redundant. This is also reported in the following publication in cells: Plouffe, S. W. et al. J. Biol. Chem., 2018, https://doi.org/10.1074/jbc.RA118.002715. Furthermore, YAP and TAZ double-knockout (KO) mice display a more severe phenotype than either of the single-KO mice (Xin, M. et al, Proc. Natl Acad. Sci. USA, 2013, https://doi.org/10.1073/pnas.1313192110; Nishioka, N. et al., Dev. Cell , 2009, DOI: 10.1016/j.devcel.2009.02.003), suggesting that there are some functional overlaps. We included the sentence and the reference in the main text (page 19, lines 1052-1055). On the other side, many of the mentioned compounds are characterized by polypharmacology (see also the response reported below).
Point 6: Finally the quinonic function could simply explain the cytotoxicity of IA-5 by production of ROS as in the case of digoxin and deslanoside.
Response 6: we thank the Reviewer for notifying us. We are aware that quinones are highly redox active molecules leading to the formation of reactive oxygen species (ROS), which can alter redox balance within cells through the formation of oxidized cellular macromolecules including lipids, proteins, and DNA. This accounts at least in part, for the reduced viability of all our cell lines after treatment with the compounds, IA-5, digoxin and deslanoside. Both deslanoside and digoxin are known to induce ROS formation (Bolton and Dunlap, Chem. Res. Toxicol. 2017, DOI: 10.1021/acs.chemrestox.6b00256; Wang et al., Front. Pharmacol. 11:186, 2020, doi: 10.3389/fphar.2020.00186). On the other hand, and in addition, it has been shown that inactivation of YAP, decreases antioxidant gene expression, leading to enhanced oxidative stress-induced cell death through downregulation of catalase and MnSOD (Shao et al., Nat Commun. 2014 doi:10.1038/ncomms4315). Therefore IA5, digoxin and deslanoside can induce in part cancer cell reduced viability through ROS production induction.
Therefore, we are grateful to the Reviewer for the suggestion of mentioning also this possible mechanism involved in the action of these compounds targeting YAP and report a brief comment in the Discussion (page 19, lines 1041-1046).
Reviewer 2 Report
In this manuscript, Lauriola et al characterize a 27-compound library and identify compounds that modulate YAP-TEAD activity. They narrow down their search to the compound IA5, a quinone derivative, as it reduces cell proliferation, YAP/TEAD-dependent reporter activity and also YAP expression levels. The p-YAP/YAP ratio appears to be increased after IA5 treatment, suggesting that the compound may also facilitate cytoplasmic localization of YAP but it is not entirely clear from the immunofluorescence data. The manuscript reads really well and the data mostly supports the authors’ conclusions. However, I have the following concerns that needs to be addressed:
1. How do the authors ascertain that decrease in cell proliferation is due to inhibition of YAP/TAZ activity, especially as the authors mention in the discussion section that IA5 is known to inhibit cell proliferation through other mechanisms. For the cell lines used in this study, including HaCaT, there is no evidence provided that indicates that cell proliferation is dependent on YAP-TEAD activity. If this is not shown, how can the authors use cell proliferation as a readout to identify YAP/TEAD inhibitors? Can the authors show whether knockdown of YAP or TEAD levels has an impact on cell proliferation in at least a couple of the cell lines used in this study. Or can they at least cite previous studies where these cell lines were used to test the effect of YAP/TEAD knockdown on cell proliferation?
2. How did the authors calculate the protein expression levels and the pYAP/YAP ratio - Figures 8C and 8D when the YAP band in Figure 8B is not visible at all? Interestingly, most of the compounds used in the study decrease YAP expression levels but their mechanism of action is not entirely clear.
3. The introduction part is nicely done but it is too long and also includes two figures. However, this gives the impression that this is a review and not a research article.
Author Response
Response to Reviewer 2
In this manuscript, Lauriola et al characterize a 27-compound library and identify compounds that modulate YAP-TEAD activity. They narrow down their search to the compound IA5, a quinone derivative, as it reduces cell proliferation, YAP/TEAD-dependent reporter activity and also YAP expression levels. The p-YAP/YAP ratio appears to be increased after IA5 treatment, suggesting that the compound may also facilitate cytoplasmic localization of YAP but it is not entirely clear from the immunofluorescence data. The manuscript reads really well and the data mostly supports the authors’ conclusions. However, I have the following concerns that needs to be addressed:
Point 1: How do the authors ascertain that decrease in cell proliferation is due to inhibition of YAP/TAZ activity, especially as the authors mention in the discussion section that IA5 is known to inhibit cell proliferation through other mechanisms. For the cell lines used in this study, including HaCaT, there is no evidence provided that indicates that cell proliferation is dependent on YAP-TEAD activity. If this is not shown, how can the authors use cell proliferation as a readout to identify YAP/TEAD inhibitors? Can the authors show whether knockdown of YAP or TEAD levels has an impact on cell proliferation in at least a couple of the cell lines used in this study. Or can they at least cite previous studies where these cell lines were used to test the effect of YAP/TEAD knockdown on cell proliferation?
Response 1: we thank the Reviewer, and we agree with the suggestion. According to Reviewer's suggestion, some references to previous studies (Lijuan Wang et al, PLoS One, 2013, doi: 10.1371/journal.pone.0065539; Meng Su et al, J Gastrointest Oncol, 2021, DOI: 10.21037/jgo-21-258) show that the cell lines reported by us were used to test the effect of YAP/TEAD knockdown on cell proliferation. We have been included those references in the Main Text (page 16, lines 935-937).
Point 2: How did the authors calculate the protein expression levels and the pYAP/YAP ratio - Figures 8C and 8D when the YAP band in Figure 8B is not visible at all? Interestingly, most of the compounds used in the study decrease YAP expression levels but their mechanism of action is not entirely clear.
Response 2: we thank the Reviewer for notifying this point. According to the journal guideline for high quality representative picture, we decided to use the blot with short exposure in the figure 8B (in the revised Manuscript replaced with Figure 7B). The quantification has been done by using blots with medium/long exposure from three independent experiments (see raw data). We agree with the Reviewer's comment and included the specific sentence in Figure legend (page 18, lines 1022-1023).
Point 3: The introduction part is nicely done but it is too long and also includes two figures. However, this gives the impression that this is a review and not a research article.
Response 3: We are thankful to the reviewer for raising this point. According to her/his suggestion, we have shortened the Introduction and moved Figure 2 in the Supplementary Material (see Figure S1).
Round 2
Reviewer 1 Report
The corrections made by the authors correspond to my recommandations.